# Causes of Hass Avocado Fruit Rejection in Preharvest, Harvest, and Packinghouse: Economic Losses and Associated Variables

**Joaquín Guillermo Ramírez-Gil [1],\*** , **Jaime Horacio López [2]** and **Juan Camilo Henao-Rojas [2]**

[1] Departamento de Agronomía, Facultad de Ciencias Agrarias, Universidad Nacional de Colombia, 111321 Sede Bogotá, Colombia

[2] Centro de Investigación La Selva-Rionegro, Corporación Colombiana de Investigación Agropecuaria-Agrosavia, 111321 Ríonegro, Colombia; jlopezh@agrosavia.co (J.H.L.); jhenao@agrosavia.co (J.C.H.-R.)

\* Correspondence: jgramireg@unal.edu.co

**Abstract:** The areas planted with avocado in Colombia have increased to position this fruit in international markets. To achieve this goal, the offered fruits need to meet optimal production standards. The aim of this study was to identify the main physiopathologies and damages that may cause the rejection of avocado cv. Hass fruits for export purposes during pre-harvest, harvest, and processing in packinghouses, and quantify the economic impact of said exclusion criteria. Typological characterization and quantification of damages that may cause fruit rejection were performed at the laboratory, field, and packinghouses. Data were obtained from 136 Colombian productive plots with monitoring and data collection records for a minimum of five years. At the packinghouse level, data associated with factors that affect quality (both, those identified at packing plants and those detected in simulated shipments) were considered. The main causes of fruit rejection during pre-harvest and harvest were: low calibers, damage to the epidermis by insect pests from the subfamily *Melolonthidae*, thrips, mites, the bug Monalonion, overripe fruits, and sunburn damage. In addition, pathologies such as anthracnose and stem end rot, and problems associated with browning of vascular bundles, irregular ripening, chilling injury, and lenticel damage were highly limiting at packinghouses. The economic analysis identified losses of 5.78 and 5.68% in farms and packinghouses, respectively, which are equivalent to US dollar (USD) 80.29 per produced ton. This work generated information that allows prioritizing strategies to improve fruit quality and reduce economic impact.

**Keywords:** quality; damages; economic losses; Hass avocado

## 1. Introduction

Avocado (*Persea americana* Mill.) is a tropical species of the family Lauraceae that includes three known races [1,2]. Different commercial varieties have originated from interspecific recombinations, which have expanded avocado's distribution from tropical climates to subtropical and Mediterranean areas [3]. The Hass variety is the most cultivated worldwide and is mainly grown in countries such as Mexico, Chile, Peru, Colombia, Indonesia, South Africa, Israel, Kenya, United States, and Brazil [4].

Avocado in Colombia, especially the Hass variety, has shown a significant increase in planted areas and production in recent years [5]. This growth is supported by a higher domestic demand and the boom in exports, especially to the European Union [6]. There is currently a great expectation for the possibility of exporting avocados to other potential markets such as Asia and the United States.

Fruit and vegetable quality translates into nutritional, microbiological, technological, commercial, and sensory indicators, and parameters associated with functional properties, origin, and defects also play an important role [7]. Avocado fruit quality is achieved in crops as a result of genetic factors, environmental conditions, agronomic management, harvest criteria, fruit selection, and pest and disease presence, among others, and must be preserved and taken to its maximum expression during post-harvest practices [7–12].

In Colombia, there is not enough information on quality parameters for avocado cv. Hass fruits for export. Only a few studies about the quantity and quality of fatty acids, fruit size, extra quality percentages, pulp yield, nutrient content and color have been reported [13–15]. Due to this situation, Colombian fruits have obtained an average score of 5.9 out of 10 in the ranking by European quality (6.1). These values were much lower than the ones obtained by direct competitors such as Israel (6.8), Peru (7.1), South Africa (7.2), and Chile (7.4) [16].

Given the little information on the causes of damage that affect the external quality of avocado cv. Hass in Colombia and the need to establish the most limiting quality parameters for avocado in the country [16], this work had three objectives. First identify the main problems that affect the quality of avocado cv. Hass for export during pre-harvest, harvest, and packinghouse. Second estimate the economic impacts of said defects on the Colombian productive chain. Third identify the relationship between environmental, topographic, and agronomics variables and incidence of rejection cause. This research was carried out based on the records of 136 plots that export avocado in different producing regions in Colombia and data supplied by six fruit packinghouses for export purposes.

## 2. Materials and Methods

### 2.1. Plant Material and Sampling

The data used in this study were based on the information obtained from records of 136 plots (including 46 plots with damage characterization) that produce avocado cv. Hass and six packinghouses from the departments of Antioquia, Caldas, Risaralda, Quindío, Tolima and Valle del Cauca. These data were collected from a minimum of 10 harvests per plot (main and secondary).

The number of plots and packinghouses was determined based on a simple random sampling, using the maximum variance formula (Equation (1)) [17]. Data provided by the Colombian Agricultural Institute (ICA) (such as crops with registration for export and trading companies with fruit shipments to foreign countries) were considered to determine the population. Additionally, plots that met the following requirements were selected: (i) size greater than 4 ha, (ii) production and export history of more than five years, and (iii) records of fruit damage and defects associated with physiopathology, insects, mechanical factors, rodents, environmental factors, nutritional deficiencies, harvest criteria, and size, among others. In addition, crops with environmental conditions within the ranges for productive systems in Colombia, with locations between 1800 and 2500 m elevation, average annual temperature between 14 and 20 °C, precipitation of 1200–2600 mm and relative humidity between 75 and 95% [5]. At the packinghouse level, data were related to rejections due to the aforementioned damages, information generated in simulated shipments and the criteria of each packing company according to the market requirements.

$$n = \frac{Z^2 \, P(1-P)/e^2}{1 + (Z^2 \, P(1-P)/e^2 \, N}$$

(1)

where:

n: sample size
N: population size
e: error range (10%)
z: value based on the confidence level (90% = 1.65)

P: probability (90%).

### 2.2. Characterization of Damage and Defects Associated with Pre-harvest, Harvest and Packinghouse Processes that Affect Avocado cv. Hass Quality and Cause Fruit Rejection

The characterization of damages that affect the external quality of Hass avocado fruits and the identification of the origin or factor involved were possible through a detailed analysis of 46 production systems over a period of eight years (2009–2016) [11]. In each plot, fruit growth, ripening, harvest, and transport were monitored. Problems associated with physiopathologies in the field and packinghouse were corroborated by a polyphasic analysis at the Laboratory of "Fitotecnia Tropical" at Universidad Nacional de Colombia, Medellín campus (Ramírez-Gil and Morales, in press). Regarding the identification of insects and mites that affect fruits, this process was carried out at the Francisco Luis Gutiérrez entomological museum (MEFLG) at Universidad National de Colombia, Medellín campus. The specific identification process of each damage factor that affects the external quality of Hass avocado fruit is described as follows:

Fruit malformations associated with boron and zinc deficiencies were identified by visual inspection of characteristic symptomatic patterns [18,19]. In the case of damage to the fruit epidermis caused by abiotic factors such as sunlight and hailstones, herbicide toxicity, ring-neck, and necrotic seed that adheres to the pulp, the descriptions of these problems reported in the literature were used as reference [20].

Regarding abiotic and biotic factors that cause disorders and pathologies that affect the external quality of avocado at preharvest, harvest, and packinghouses such as chilling injury, fruit affected by impacts, lenticel damage, stem end rot, anthracnose, surface cracks, scabs, and epidermis rot, their patterns were corroborated with the information reported for each of these causes of damage [10, 21–25]. Additionally, for pathologies of biotic origin, isolation of the causal agents was performed. For *Phytophthora* spp. vegetable juice culture medium (180 mL $L^{-1}$), with additions of agar (24 g $L^{-1}$ Difco, Franklin Lakes, Nueva Jersey, NJ, USA), ampicillin ( 200 μg $L^{-1}$), chloramphenicol (20 μg $L^{-1}$), and benlate (Benomyl®, 100 μg $L^{-1}$) (V8-AACB) were used; potato dextrose agar acidified with lactic acid (PDA-A) (Difco, Franklin Lakes, Nueva Jersey, NJ, USA) for fungi and nutritive agar (NA) (Difco, Franklin Lakes, Nueva Jersey, NJ, USA) supplemented with fungicide (Benomyl® 50 μg $L^{-1}$) for bacteria. Morphological characterization of fungi was performed using the keys by Barnett and Hunter [26]. Oomycete characterization was carried out using the keys by Erwin and Ribeiro [27], and molecular characterization was performed by amplification and sequencing of ITS regions 1, 4, and 5 [28]. For the particular case of the isolated bacteria, basic biochemical tests were used [29]. For each of these microorganisms, their pathogenicity on avocado fruits was confirmed.

For postharvest associated problems identified at packinghouses, simulated shipments were performed using fruit from different plots. For these simulations, a 90% relative humidity and a temperature of 5 °C were established, and oxygen, carbon dioxide, and ethylene concentrations were controlled by automated air relays and the use of ethylene absorbers in the boxes (8 g sachet). A simulation was carried out for a period of 30 days, according to the times used in the Colombian fruit transport chain to European markets.

For damage caused by insects or arthropods, their identification and characterization was performed by using two approaches. The first one followed the characteristic reports that describe damage in the field caused by insect and arthropod pests that affect Hass avocado fruits and how they compromise quality under Colombian tropical conditions [30–33]. The second approach consisted in the manual capture or through the use of entomological nets of individuals in plots with high levels of insect damage and/or populations on affected structures. These individuals were placed in jars with a 70% alcohol-sterile distilled water solution (*v:v*) and transported to the MELG for identification at the genus level using taxonomic keys [30,34–40].

In the case of the damages related to malformation and lumps, changes in the epidermis color, rounded fruits, pedicel detachment and rodents, these were identified according to the characteristics

of the damage and the origin was determined according to interviews and reports of technical assistants at the evaluated plots. In the case of fruit size, fruit weight was used as exclusion criterion and fruits that weighed less than 90 g were rejected [11]. Overripe fruit was associated with changes in firmness (pre-conditioned, breaking, and firm ripe), loss of brightness and a purple to deep purple color equivalent to the color scale between 2 and 4 according to Ochoa, [10].

Finally, from the data obtained in the 46 plots, a detailed guide was made including the description of symptoms and the factor or factors involved in the damages that affect the external quality of fruits, which was shared and discussed with the technical assistants of the remaining 90 plots. With this guide, a standardization of damage reports for each crop was achieved, eliminating those in which damage expression did not coincide. In addition, confusing reports with very low frequencies of occurrence or without damage characterization were not taken into account.

### 2.3. Economic Importance Associated with Damages and Defects in Farms and Packinghouses that Generate Fruit Selection and Rejection Criteria

The economic importance of rejected fruit was determined through a multi-step analysis [11]. This method determines the production costs of avocado crops based on their stage of development, technological level, and production region to which they belong. In this particular case, the following procedure was used: (i) determination of the technological level and location of the 136 evaluated plots, (ii) production costs of one kilogram of fruit in the 136 evaluated plots according to the technological level and production region; and (iii) determination of the economic value of fruit not sold due to rejection factors in the field, harvest, and packinghouse according to Equations (2) and (3). Losses were expressed in dollars per ton of produced fruit, based on an exchange rate of 1 USD: 2930 COP for Colombia.

$$\text{PTD} = \sum_{i}^{n} (\text{Pi}/\text{PT}) \times 100 \tag{2}$$

where:

PTD: total losses due to rejected fruit in %.
P: production losses due to rejection caused by factor i that affects external quality.
PT: total production of the plot according to the technological level and region where it is located.
n: amount of damage generated by fruit rejection.
i: specific damage factor that affects the external quality of fruits and generates rejection.

$$\text{IEPD} = \text{PTDpc} \times (\text{CP}) + \text{PTDe} \times (\text{CP} + \text{CT} + \text{CM}) \tag{3}$$

where:

IEPD: economic importance of losses by fruit rejection.
PTDpc: total fruit losses due to pre-harvest and harvest rejection.
PTDe: total fruit losses due to packinghouse rejection.
CP: fruit production costs according to the technological level and location of the plot.
CT: transport cost from plot to packinghouse.
CM: cost of fruit processing in packinghouse.

### 2.4. Data Analysis

The incidence, origin of damage and rejection of fruits in each production system evaluated was analyzed by means of descriptive statistics using the population mean and calculating the standard deviation. All the analyses performed in this study, including the determination of the sampling unit, descriptive analysis, characterization of technological levels, losses in production and economic impact associated with rejected fruits, were carried out through the implementation of these functions in the statistical program R version 3.6.0 [41].



*2.5. Relation between Causes of Rejection and Environmental and Management Variables*

The environmental variables included in the evaluation were temperature (°C), precipitation (mm), and relative humidity (%), obtained from weather stations (WatchDog serie 2000™, Spectrum Technologies, Aurora, CO, USA) located in some of the plots evaluated. Environmental data for the plots where no stations were available, were obtained by interpolation using data from the climatic network available at Ideam, Colombia (Institute of hydrology, meteorology and environmental studies), using the Higher Order Delaunay Triangulations (HODTs) [42], implemented in R [41]. These variables were complemented with elevation data obtained for each avocado plot using a GPS equipment (Triple Serie GeoXT, Trimble Navigation Limited, Westminster, CO, USA) adjusted under the projection system UTM WGS 84 Zone 18 N. In addition, each plot was classified in a technological level as low, medium, and high [11].

In order to identify if there are clusters between the environmental, topographic and management variables and the incidence of each of the rejection causes, unsupervised and supervised grouping tools were used. This process was carried out by integrating K-means and support vector machine algorithms implemented in R [41]. These methods were calibrated with 70% of the data and their validation was carried out with 30% of the remaining data by the cross-validation method.

## 3. Results

*3.1. General Description of Damages Associated with the External Quality of Hass Avocado Fruits*

Avocado fruits and their external quality in the 136 evaluated plots and packinghouses were altered by different factors, which were grouped into four clusters. In a first group, those alterations that affected fruit shape and size were identified. Some of the factors reported in this group are: malformations or lumps with unknown origin (Figure 1A), boron (Figure 1B), and zinc (Figure 1C) deficiencies, very rounded fruits with unknown origin (Figure 1E), fruits with reduced size and necrotic seed that adheres to the pulp, whose origin seems to be related to periods of fruit filling with water deficit as a consequence of the "El niño" phenomenon (Figure 1P), and fruit malformation due to the avocado ovary gall midge (*Bruggmanniella perseae,* Gagné) (Figure 2E,F).

Another group of defects that affected quality was associated with factors that alter the fruit epidermis and, to a lesser extent, the pulp. Within these factors we could find changes in the epidermis color with unknown origin (Figure 1D), damage caused by mechanical tools (Figure 1F), black and brown spots caused by herbicide drift and sunburn (Figure 1H,K), breaking of the epidermis by hailstones (Figure 1L), and lenticel damage (Figure 1Q). This cluster also grouped damage caused by the majority of insects and arthropods that alter the epidermis, cause lumps, induce exudates on lesions, or simply settle on the tissue. Within these arthropods there are mites (Figure 2A) (*Olygonichus yothersi* Mcgregor), the Melolonthidae complex (Figure 2B) (*Astaena pigydialis* Kirsch, mainly), thrips (Figure 2C) (Thysanoptera), the bug monalonion (Figure 2D) (*Monaloniun velezangeli* Carvalho and Costa), and scales (Figure 3G,H) (*Hemiptera: Coccoidea*). Similarly, different rodents that cause epidermis detachment and holes in the pulp were included in this cluster (Figure 3N,O).

Physiopathologies make up another group of fruit rejection causes with a strong presence mainly during postharvest. Some of the pathologies of abiotic origin found were the appearance of necrotic areas in the epidermis and pulp associated with impacts (Figure 1J), irregular ripening characterized by pulp areas with different ripening levels, adhered and pre-germinated seeds (Figure 1M–O) and chilling injuries with different expressions which include epidermis areas with intense black coloration and pulp and fibers with browning and purple coloration (Figure 2R). On the other hand, the pathologies of biotic origin were anthracnose (*Colletotrichum gloeosporioides sensu lato*), stem end rot (*Rhyzopus* sp., *Lasiodiplodia* sp., *Pestalotia* sp., and *Phomopsis* sp.), avocado scab (*Sphaceloma persea* Jenk), cracks in the epidermis caused by Cercospora spot (*Pseudocercospora purpurea* (Cooke) Deighton), pedicel lesions (*Colletotrichum gloeosporioides sensu lato*, *P. purpurea*, *S. persea*, *Pestalotia* sp.) and fruit

epidermis rot (*Rhyzopus* sp., *Lasiodiplodia* sp., *Botryosphaeria obtusa* (Schwein), *Dothiorella* sp., *Pestalotia* sp., and *Phomopsis* sp.) (Figure 3).

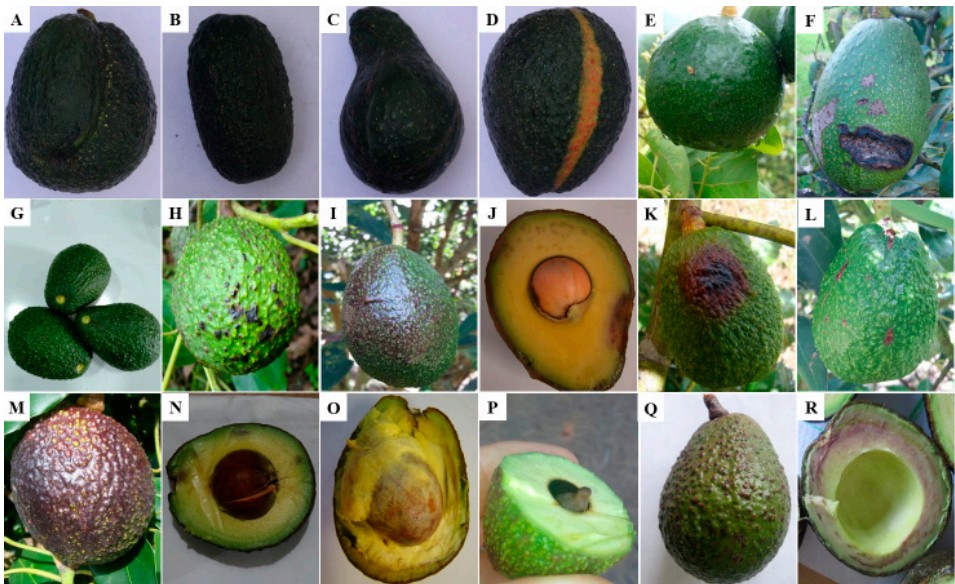

**Figure 1.** Malformation, nutritional deficiencies, mechanical defects, disorders and environmental damage affecting fruit in Hass avocado: (**A**) malformation of unknown origin; (**B**) boron deficiency; (**C**) zinc deficiency; (**D**) natural variegation; (**E**) fruit shape; (**F**) mechanical damage; (**G**) fruit without pedicel; (**H**) herbicide damage; (**I**) ring-neck; (**J**) impact damage; (**K**) sunburn damage; (**L**) hailstone damage; (**M–O**) overripe fruit; (**P**) necrotic seed; (**Q**) lenticel damage; (**R**) chilling injury.

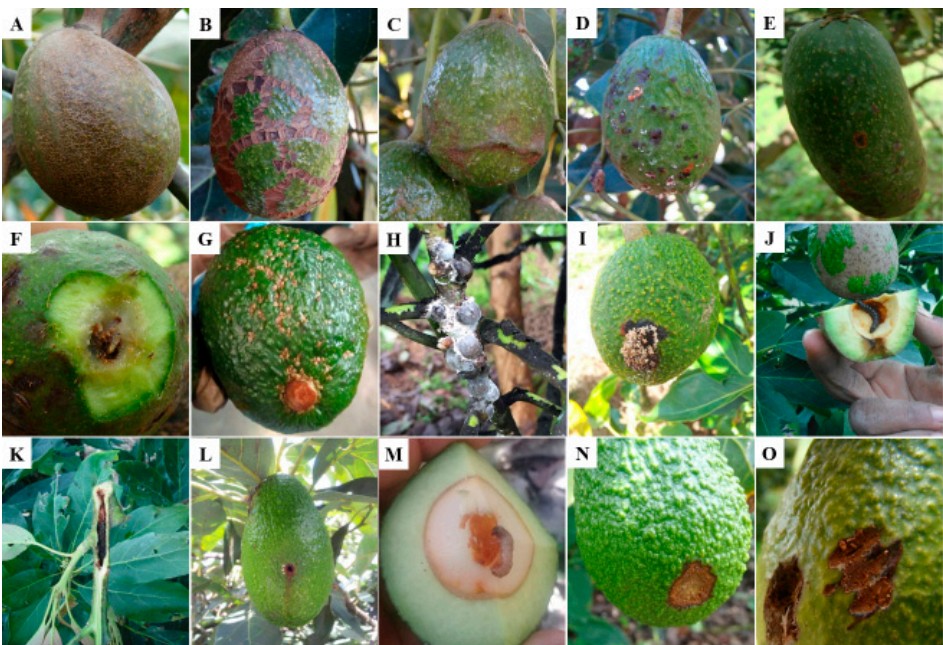

**Figure 2.** Damage caused by pests affecting fruit in Hass avocado: (**A**) mites (principally associated with *Olygonichus yothersi* Mcgregor); (**B**) Melolonthidae (principally associated with *Astaena pigydialis* Kirsch); (**C**) thrips (Thysanoptera); (**D**) the bug monalonion (*Monaloniun velezangeli* Carvalho and Costa); (**E,F**) ovary gall midge (*Bruggmanniella perseae* Gagné); (**G,H**) scales (*Hemiptera: Coccoidea*) (Photos: Wilmar Perez); (**I–K**) avocado seed moth (*Stenoma catenifer* Walsingham) (**L,M**) big avocado seed weevil (*Heilipus lauri* Boheman). (**N,O**) rodent damage.

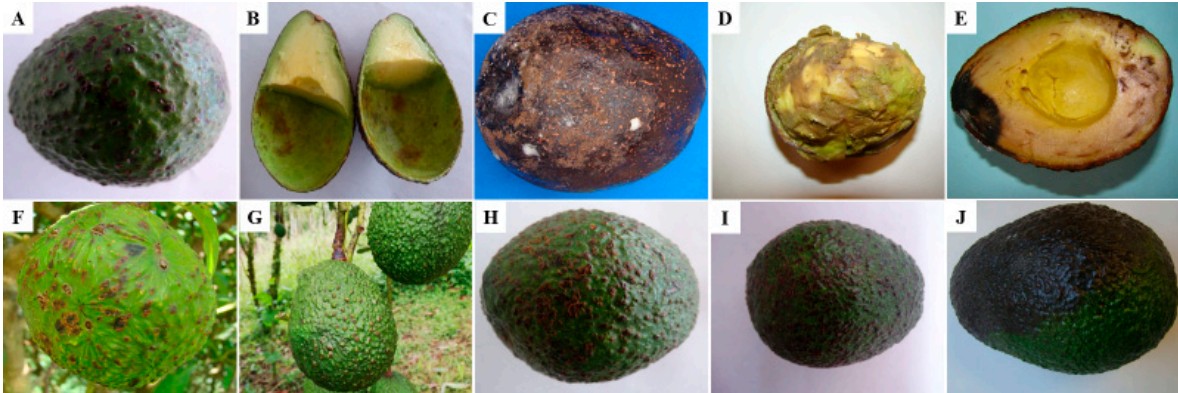

**Figure 3.** Symptoms of disease affectation on fruit in Hass avocado, A to D symptoms associated with anthracnose (*Colletotrichum gloeosporioides sensu lato*). (**A**) lesion in external epidermis; (**B**) lesion in internal epidermis; (**C**) advanced infection in external epidermis; (**D**) advanced infection in all mesocarp; (**E**) typical symptoms associated with stem end rot caused by different microorganisms (*Rhyzopus* sp., *Lasiodiplodia* sp., *Dothiorella* sp., *Pestalotia* sp., and *Phomopsis* sp.); (**F**) superficial cracks caused by *Pseudocercospora purpurea* (Cooke) Deighton; (**G**) pedicel lesion caused by different microorganisms (*Colletotrichum gloeosporioides sensu lato*, *Pseudocercospora purpurea* (Cooke) Deighton, *Sphaceloma persea* Jenk, *Pestalotia* sp.) (Photo: Anibal Lopez); (**H**) scab caused by *Sphaceloma persea* Jenk; (**I**) simultaneous infection caused by *Colletotrichum gloeosporioides sensu lato* and *Sphaceloma persea* Jenk; (**J**) fruit rot caused by different microorganisms (*Rhyzopus* sp., *Lasiodiplodia* sp., *Botryosphaeria obtusa* Schwein, *Dothiorella* sp., *Pestalotia* sp., and *Phomopsis* sp.). Causal agents and symptoms for each of the pathologies were based on reports by Ramírez-Gil (2018).

In the last group of damages that cause immediate fruit discard we could find fruit without pedicel (Figure 1G), ring-neck (Figure 1I), overripe fruit (Figure 1M), and fruit with damage caused by quarantine pests such as *Stenoma catenifer* Walsingham and *Heilipus lauri* Boheman (Figure 2I–M).

*3.2. Incidence and Rejection Associated with the Main Damages in Pre-harvest, Harvest and Packinghouse that Affect Hass Avocado Quality*

Fruit damages or alterations related to malformations, changes in color, nutritional deficiencies, impacts, environmental conditions (hailstones, sunlight) and agronomic practices (damage by machinery) presented different degrees of relevance, both at field and packinghouse levels. Fruit malformations (Figure 1A) and changes in the epidermis color (Figure 1D) showed minimal incidence, and no rejection data were reported. Regarding boron and zinc deficiencies (Figure 1B,C) their incidence was less than 0.3%, with rejection values greater than 90% (Figures 4 and 5A). The following fruit discard causes showed incidences below 0.5% (Figure 4): presence of very rounded fruit shape, with rejection values between 35% and 40% in the field and packinghouse (Figures 1E and 5A); damage caused by mechanical tools (Figures 1F and 6A), fruit without pedicel (Figures 1G and 5A), herbicide phytotoxicity (Figures 1H and 5A), overripe fruit (Figure 1M–O and Figure 5A), and necrotic seed that adheres to the pulp (Figure 1P) all had 100% rejection; finally, epidermis damage due to hailstones had rejection values between 15% and 35% in the field and packinghouse (Figures 1L and 5A).

Ring-neck (Figure 1I), lenticel damage (Figure 1Q) and fruit size were the most relevant fruit discard causes especially during pre-harvest, with incidence and rejection values of 0.4–100%, 1–15%, and 3.6–50%, respectively (Figures 4 and 5A). Regarding post-harvest, fruit quality was altered by impacts (Figure 1J), lenticel damage (Figure 1Q), chilling injury (Figure 1R), fruit size and sunburn damage (Figure 1K), with incidences of 0.8%, 2.4%, 1.8%, 1.5%, and 0.8%, respectively, and different rejection percentages (between 20% and 80%) (Figure 4).

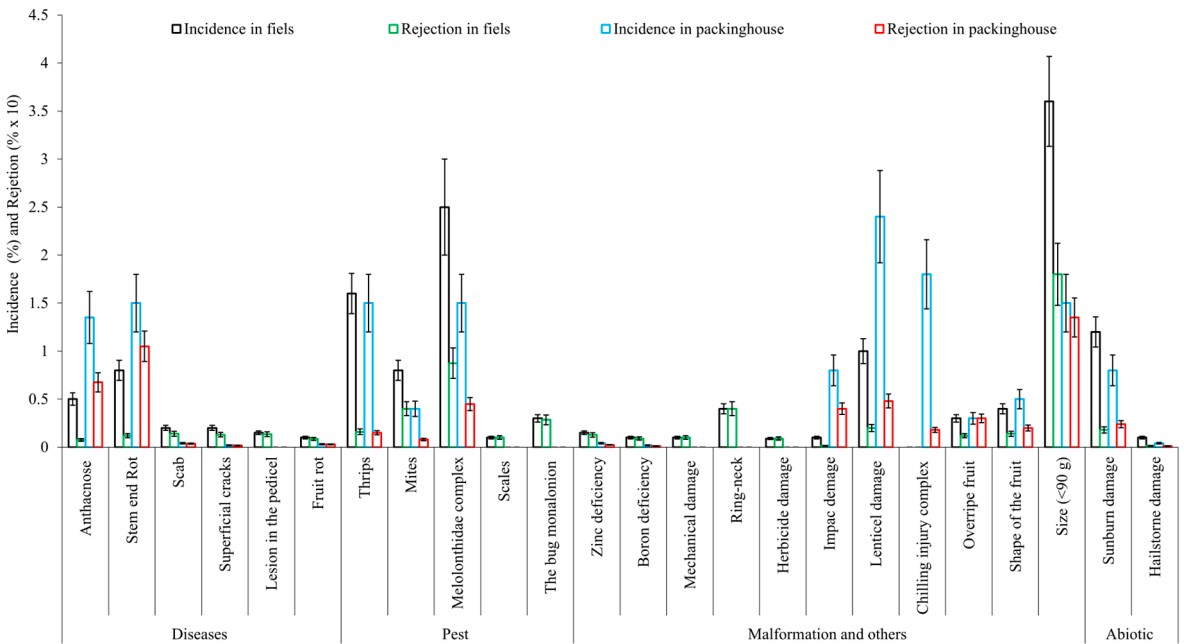

**Figure 4.** Incidence and rejection of avocado Hass fruits affected with diseases, pests, malformation, mechanical damage, disorders, and abiotic damage. Data from fields were obtained from 136 plots planted in the departments of Antioquia, Caldas, Risaralda, Quindío, Valle, and Tolima. Data from packinghouses were obtained from four companies. Line on bars represents the standard deviation of data.

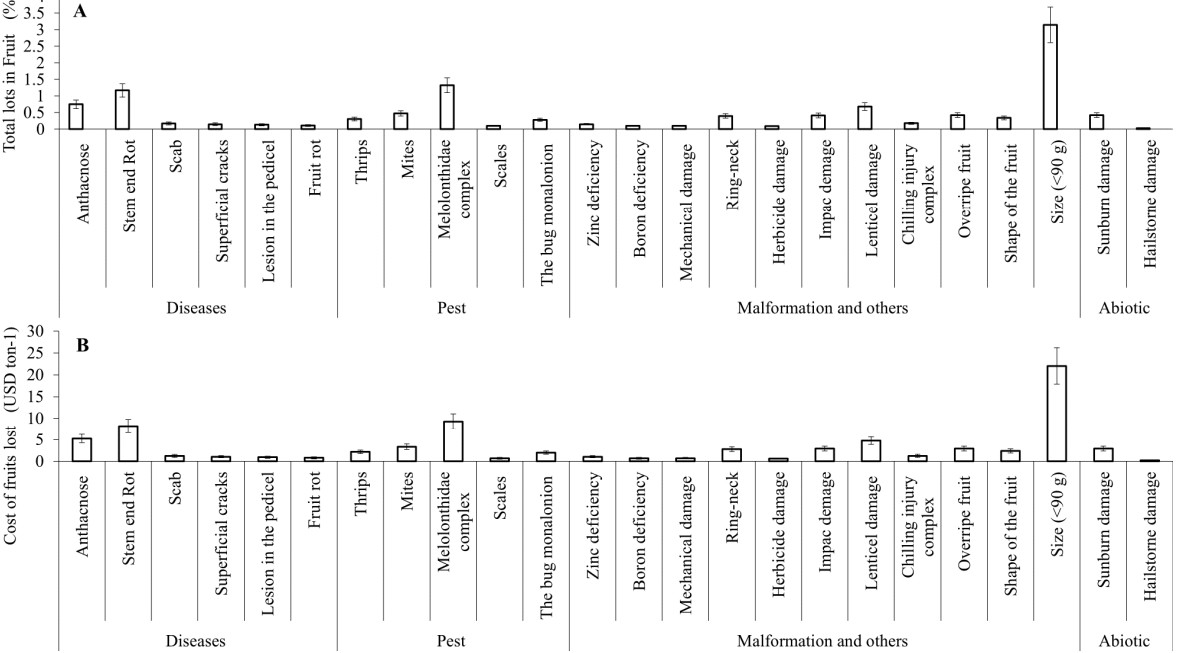

**Figure 5.** Economics loss and costs of avocado Hass fruits associated with damage caused by diseases, pests, malformation, mechanical factors, disorders, and abiotic problems. (**A**) data is the sum of fruit rejected in fields and packinghouses; (**B**) the cost was obtained based on Equation (1). Data from fields were obtained from 200 plots planted in the departments of Antioquia, Caldas, Risaralda, Quindío, Valle, and Tolima. Data from packinghouses were obtained from four companies. Line on bars represents the standard deviation of data. Losses were expressed in dollars per ton of produced fruit, based on an exchange rate of 1 USD: 2930 COP for Colombia.

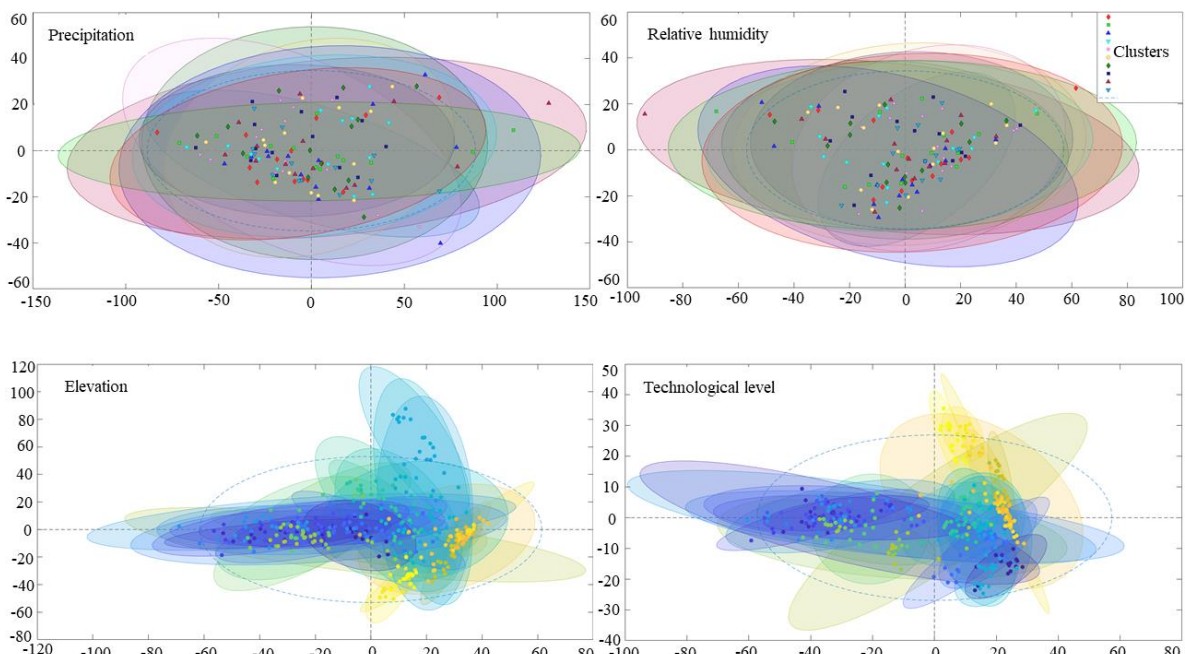

**Figure 6.** Supervised and unsupervised clustering of the incidences of avocado rejection and environmental, topographic, and crop management variables. Clusters were made using K-means and support vector machine algorithms.

In the case of damages associated with insects and rodents, their relevance was not strictly related to incidence, but with the level of fruit alteration and the restriction that some pests have as quarantine pests. Thus, mites (Figure 2A), the *Melolonthidae* complex (Figure 2B) and thrips (Figure 2C) have the highest incidence with values between 1 and 2.5% (Figure 5) and rejection in the field and packinghouse of 20–50%, 20–35%, and 10–15.6%, respectively. Packinghouses tolerate these damages so their discard is low. Regarding damages caused by the bug monalonion (Figure 3D) and scales (Figure 2G,H), their incidences were less than 0.3% in the field, with 100% rejection in the field and packinghouse since their damage is not tolerated (Figures 4 and 5A). On the other hand, data regarding ovary gall midge (Figure 2E,F) and rodents (Figure 2N,O) incidence in the field were not taken into account since they appear very sporadically and fruit was immediately discarded when they were spotted. In the case of pests with quarantine restrictions, the presence of *Stenoma catenifer* Walsingham (Figure 2I–K) and *Heilipus lauri* Boheman (Figure L,M) was sporadic and fruits and branches were removed when they were spotted in a plot.

Pathologies such as anthracnose (Figure 3A–D) and stem end rot (Figure 3E) were highlighted in the plots studied in this work. For these pathologies, incidence and rejections of affected fruit at packinghouses showed values of 1.3–52.5% and 1.5–70.1%, respectively, which were much higher than those reported in the field (0.5–15.3% and 0.8%-18.5%, respectively) (Figures 4 and 5A). Other pathologies with lower incidence were: avocado scab, cracks in the epidermis, pedicel lesions, and fruit rot with incidences below 0.2%, and rejection values greater than 65 and 90% in the field and packinghouse (Figures 4 and 5A).

### 3.3. Economic Importance of Damages and Defects in Farms and Packinghouses that Generate Fruit Selection and Rejection Criteria

The costs of total fruit discarded for different reasons (Figures 4 and 5A) in farms (5.78%) and packinghouses (5.68%) were USD 80.29 per each ton of produced fruit (Figure 6B). The following losses are highlighted: fruit size (USD 22), the Melolonthidae complex (USD 9.25), stem end rot (USD 8.19), anthracnose (USD 5.25), lenticellosis (USD 4.76), fruit damage caused by mites (USD 3.36),

overripe fruit (USD 2.94), sunburn damage (USD 2.93), fruit affected by impacts (USD 2.90), ring-neck (USD 2.8), rounded fruit shape (USD 2.38) and thrips damage (2.17 USD) (Figure 5B). For the other problems affecting fruit quality, the economic value was less than 2 USD (Figure 5B).

*3.4. Relation between Causes of Rejection and Environmental and Management Variables*

The methods of unsupervised and supervised grouping showed that the incidences of the causes of rejection of the fruit in avocado are highly influenced by the elevation and the technological degree of the production system, while the other variables did not allow to identify groups (Figure 6).

## 4. Discussion

Avocado fruits and their quality can be altered by different factors. The presence of certain damage can only be cosmetic or can compromise fruit quality in terms of nutritional and sensory contents. In addition, it can threaten the phytosanitary safety of the importing country. Fruit rejection or inclusion for export purposes is determined based on damage intensity and type, and each trading company and importing country has different exclusion or damage tolerance criteria.

Avocado fruit quality is the sum of the entire production process, which involves plant material suppliers, producers, packers, distributors, and consumers. The synergistic relations between each of these links in the production chain are acquiring more importance day by day [43]. That is why it is necessary to establish clear communication channels in order to define quality standards and develop sufficient technologies to increase quality and reduce economic losses due to discarded fruit.

Some of the quality-related problems found in this study are inappropriate harvest and post-harvest practices since the mechanical damage generated at these stages affects the cosmetic appearance of fruits or may be the gateway to pathogens that cause postharvest diseases. Mechanical damage can also generate higher metabolic rates in fruits, and therefore, premature and irregular ripening [7,12]. In addition, harvest under appropriate environmental and logistic conditions that meet the right criteria or standards (harvest index, dry fruit, packaging, transport, etc.) will avoid problems related to lenticellosis, high respiration rates, disease incidence, vascular browning, among others [21,25,44].

One of the most relevant parameters that affect fruit quality identified in this study was associated with fruit ripening for consumption. It is evident that some of the most important challenges the Colombian Hass avocado industry has is offering consumers a healthy product with homogeneous ripening [15,45]. This is due to the fact that avocado is a climacteric fruit whose consumption ripening is achieved several days after its harvest [46] and its final quality depends largely on its pre-harvest conditions [24,47,48]. This situation implies that the Colombian production system still lacks information to identify the harvest criteria with the greatest influence on avocado ripening processes and their effect on fruit quality and international market perception.

Damages caused by pathologies that affect fruit external and internal quality proved to be quite limiting since their infection can occur during pre-harvest and harvest. Given their endophytic nature, disinfection and post-harvest treatments are difficult to carry out, despite the fact that this type of risk can be reduced by performing an adequate ripening process (control of temperature and relative humidity) to avoid proliferation of opportunistic organisms in fruits at consumption ripeness [24,49,50].

It was found that the causes of avocado cv. Hass fruit rejection are highly limiting in economic terms, leading to significant losses that affect the profitability of the production system. This implies the need to apply management and control measures to reduce causal factors affecting quality. The unsupervised and supervised group managed to identify how elevation is a parameter that has a high influence on the presence of many causes of rejection. On the other hand, the technological degree achieved a separation in groups depending on the level of incidence of the problems (high, high, medium, and low incidence).

**Author Contributions:** Conceptualization, J.C.H.-R., J.H.L., and J.G.R.-G.; data acquisition, J.G.R.-G.; data analysis, J.G.R.-G., and J.C.H.-R.; design of methodology, J.G.R.-G.; writing and editing, J.C.H.-R., J.H.L., and J.G.R.-G. All authors have read and agreed to the published version of the manuscript.

**Funding:** This research received no external funding.

**Acknowledgments:** The authors would like to thank the farm owners that allowed the collection of information associated with the problems that affect fruit quality and their economic importance. We also thank the avocado technical assistants who answered our call and were willing to help unselfishly with this work by sharing the farms data and their experiences. We would also like to acknowledge the packinghouses that, for commercial reasons, requested their names not to be included in the manuscript. This work not was financed by a specific institution, and the cost was assumed by the authors.

**Conflicts of Interest:** The authors declare no conflict of interest.

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
