# Peer review of "Causes of Hass Avocado Fruit Rejection in Preharvest, Harvest, and Packinghouse: Economic Losses and Associated Variables"

_agronomy, doi:10.3390/agronomy10010008_

Round 1

Reviewer 1 Report

After careful reading of the ms I found it suitable for publication, since the authors report about a large number of causes of avocado fruits rejections. The most important for me is that the results are based on so many different sources (plots), what shows their objectivity. I strongly suggest the publication of the ms after minor revison, since I found the following places in the text, which must be improved:

line 35: add "." after "Mill"

line 113: add "." after "spp"

line 220: Melolonthidae must be written with normal letters (not with italic)

line 221: replace "Thripidae" with "Thysanoptera" (also in other places in the paper), which must be written with normal letters

line 313: replace "Melonthiodae" with "Melolonthidae", which must be written with normal letters 

Author Response

Dear reviewers, we attach all the suggested corrections. We appreciate your excellent input to improve the quality of the manuscript.

Reviewer 2 Report

 The artical is with very interestic and grabbing tematic. Congratolation for that and for very hard work on it. The artical is very well constructed and aranged. 

My notes are fallow:

Line 40: should be spaced... Brazil [4] instead of Brazil[4].

Lines 57-65: should be better to transform the long sentence into some sentences. should be example bellow:

Given the little information on the causes of damage that affect the external quality of avocado cv. Hass in Colombia and the need to establish the most limiting quality parameters for avocado in the country [16], this work had three objectives. First (i) objective is to be identify the main problems...Second (ii) is to be estimate the economic impacts...

and so on..this is exapmle sentence construction.

Lines 75-82: The sentence is too long should be reduced into 2 or 3 sentences. It is too complicated for reading.

Lines 91-93: put space after 1.2. like example below

1.2. Characterization of damage and defects associated with pre-harvest, harvest and packinghouse processes that affect avocado cv. Hass quality and cause fruit rejection

 The characterization of damages that affect the external quality of Hass avocado fruits and the...

Line 116: (Benomyl® 50 µg/L) should be (Benomyl® 50 µg L-1) I thing.

Line 125: I thing the sigh of Celzius should be consite  5ºC . I saw different sign there.

Line 194:  R [41].. remove the second point

Line 294: ..(Figure 3 D) ..should be ...(Figure 3D)... NON SPACED like another figure numbers..

Line 366: ...ripeness[24,49,50]... should be ...ripeness [24,49,50]... put space before brecket.

For all equations please use the identical writing styles and better the word equation functions. Some of the letters in the equations are in italic, check it twice. 

The space before and after paragraphs are not identical used. Please check it. Specifically before and after the 1.1 to 1.5. and so on

Regards

Author Response

(The authors gave the same response as above.)
